

# Competitive effects of the macroalga *Caulerpa taxifolia* on key physiological processes in the scleractinian coral *Turbinaria peltata* under thermal stress

JianRong Fu[1,2], Jie Zhou[1], JiaLi Zhou[1], YanPing Zhang[1] and Li Liu[1]

[1] Fisheries College, Guangdong Ocean University, Zhanjiang, Guangdong, China
[2] Shanghai Engineering Research Center of Hadal Science and Technology, College of Marine Sciences, Shanghai Ocean University, Shanghai, China

## ABSTRACT

An increased abundance of macroalgae has been observed in coral reefs damaged by climate change and local environmental stressors. Macroalgae have a sublethal effect on corals that includes the inhibition of their growth, development, and reproduction. Thus, this study explored the effects of the macroalga, *Caulerpa taxifolia*, on the massive coral, *Turbinaria peltata*, under thermal stress. We compared the responses of the corals' water-meditated interaction with algae (the co-occurrence group) and those in direct contact with algae at two temperatures. The results show that after co-culturing with *C. taxifolia* for 28 days, the density content of the dinoflagellate endosymbionts was significantly influenced by the presence of *C. taxifolia* at ambient temperature (27 °C), from $1.3 \times 10^6$ cells cm$^{-2}$ in control group to $0.95 \times 10^6$ cells cm$^{-2}$ in the co-occurrence group and to $0.89 \times 10^6$ cells cm$^{-2}$ in the direct contact group. The chlorophyll *a* concentration only differed significantly between the control and the direct contact group at 27 °C. The protein content of *T. peltata* decreased by 37.2% in the co-occurrence group and 49.0% in the direct contact group compared to the control group. Meanwhile, the growth rate of *T. peltata* decreased by 57.7% in the co-occurrence group and 65.5% in the direct contact group compared to the control group. The activity of the antioxidant enzymes significantly increased, and there was a stronger effect of direct coral contact with *C. taxifolia* than the co-occurrence group. At 30 °C, the endosymbiont density, chlorophyll *a* content, and growth rate of *T. peltata* significantly decreased compared to the control temperature; the same pattern was seen in the increase in antioxidant enzyme activity. Additionally, when the coral was co-cultured with macroalgae at 30 °C, there was no significant decrease in the density or chlorophyll *a* content of the endosymbiont compared to the control. However, the interaction of macroalgae and elevated temperature was evident in the feeding rate, protein content, superoxide dismutase (SOD), and catalase (CAT) activity compared to the control group. The direct contact of the coral with macroalga had a greater impact than water-meditated interactions. Hence, the competition between coral and macroalga may be more intense under thermal stress.

Corresponding author
Li Liu, zjouliuli@163.com

## INTRODUCTION

The recent effects of climate change and other anthropogenic impacts have caused the severe degradation of coral reefs worldwide (*Leggat et al., 2022*). The first mass coral bleaching event was observed in 1998 and it killed approximately 8% of the world's coral; an additional 14% of corals were lost between 2009 and 2018 (*Souter et al., 2021*).

Many studies have asserted that ocean warming is a major factor in the reduction of coral cover (*Hughes et al., 2017*, *2019*; *Lough, Anderson & Hughes, 2018*; *Leggat et al., 2022*). For example, the successive bleaching events in 2016–2017 devastated Australia's Great Barrier Reef and resulted in an 89% decline in larval recruitment in 2018 compared to historical levels (*Hughes et al., 2017*, *2019*; *Lough, Anderson & Hughes, 2018*). During this period, 31% of reefs experienced 8–16 degree heating weeks (DHWs, °C-weeks). A decline in coral cover may lead to an increase in the cover of other benthic organisms in the reefs, such as macroalgae (*Fulton et al., 2019*). The impact of herbivorous fishes is mostly ignored, although some studies assert that overfishing and nutrient pollution is the main cause of phase shifts towards macroalgae (*Barott et al., 2012*). Research had shown that prior to 2011, the estimated global average cover of algae was low (~16%) and stable for 30 years. Since 2011, the amount of algae on the world's coral reefs has increased by about 20% (*Souter et al., 2021*). Thus, the coral reef ecosystem is undergoing an ecological phase transition to that of an ecosystem dominated by macroalgae.

Macroalgae are functional communities that are important for stabilizing reef structure (*Fulton et al., 2019*), generating primary productivity (*Fulton et al., 2014*, *2019*), maintaining nutrient cycling in reef areas, and providing food sources for herbivores (*Dubinsky & Stambler, 2011*). However, there is competition between macroalgae and corals. Macrolgae harm corals through direct contact (*Coyer et al., 1993*; *Manikandan et al., 2021*) and allelopathy (*Bonaldo & Hay, 2014*; *Fong et al., 2020*), weakening the photosynthetic performance of symbiodiniaceae (*Rasher et al., 2011*), causing the retraction of polyps (*Jompa & Mccook, 2003*), increasing the number of pathogenic microorganisms (*Clements et al., 2020*; *Rasher & Hay, 2010*), triggering coral bleaching (*Bonaldo & Hay, 2014*), and resulting in the reduced calcification and coral growth, fecundity, survival rate, and settlement rate (*Fong et al., 2020*; *Tanner, 1995*; *Leong et al., 2018*; *Rasher & Hay, 2010*). Specifically, macroalgae affect coral feeding, endosymbiont function, tissue recovery, and oxidative stress response. *Morrow & Carpenter (2008)* found that *Dictyopteris undulata* weakened the particle capture rates of *Corynactis californica* by redirecting particles around polyps and causing contraction of the feeding tentacles. The dissolved organic carbon (DOC) and terpenoids released by macroalgae decreased photosynthesis and the density of endosymbionts (*Rasher et al., 2011*; *Smith et al., 2006*; *Diaz-Pulido & Barrón, 2020*). *Bender, Diaz-Pulido & Dove (2012)* asserted that the green filamentous macroalga, *Chlorodesmis fastigiate*, significantly reduced tissue recovery in *Acropora pulchra* and led to the infection of *A. pulchra* with ciliates. High levels of reactive

oxygen species (ROS, a toxic byproduct of biological aerobic metabolism) could cause damage to cells (*Blanckaert et al., 2021*). *Shearer et al. (2012)* found that the oxidative stress response of *Acropora millepora* was activated in response to ROS by altering the transcription factors after contact with the macroalga *Chlorodesmis fastigiata* and its hydrophobic extract over a short-term period (1 and 24 h). The oxidative imbalance results in rapid protein degradation and eventually to apoptosis and/or necrosis when compensatory transcriptional action by the coral holobiont insufficiently mitigates damage.

In addition, the combined effects of ocean warming, acidification, and macroalgae contact could significantly alter the physiological response of corals (*Chadwick & Morrow, 2011*; *Kornder, Riegl & Figueiredo, 2018*; *Brown et al., 2019*; *Rölfer et al., 2021*). *Rölfer et al. (2021)* have shown that light enhanced calcification (LEC) rates of *Porites lobata* were negatively affected after contact with *Chlorodesmis fastigiata* in an ocean warming and acidification scenario, compared to coral under ambient conditions. Typically, the coral-algal competition is related to seasonal and temporal cycles. These, in turn, may be related to the abundance, biomass, and composition of macroalgae, as well as the seasonal dynamics of temperature, $pCO_2$, and light intensity (*Brown et al., 2019*, *2020*). The sensitivity of various macroalgae to environmental stressors is also different. For example, intermediate levels of ocean warming could enhance the growth and production of *Laurencia* sp. and *Lobophora* sp., which was not the case for *Sargassum* sp. (*Fulton et al., 2014*; *Hernández et al., 2018*). Additionally, overfishing and eutrophication have been shown to lead to an increased growth rate of some kinds of macroalgae (*Lapointe & Bedford, 2010*), which may indirectly enhance the competitive ability of macroalgae. Therefore, to better understand the resilience of coral reef ecosystems in the future, it is necessary to determine how coral-algal interactions are impacted by global and local stressors.

According to the China Ocean Climate Monitoring bulletin (www.oceanguide.org.cn), the average sea surface temperature (SST) in the Xuwen Sea area was 27–30 °C from May to September in 2020. In 2021–2022, the average SST that caused coral bleaching in the Great Barrier Reef from December to April was 28–30 °C (*Spady et al., 2022*). It is plausible that the physiological responses of corals in the Xuwen Coral Reef National Nature Reserve of China may be affected by thermal stress when SST reaches 30 °C. During thermal stress, coral feeding rates are drastically reduced and more energy is needed to maintain biological processes (DNA repair *etc*.,) to resist heat stress (*Ferrier-Pagès et al., 2010*; *Chakravarti et al., 2020*). Triggered by thermal stress, ROS may be produced by the endosymbionts mainly due to PS II dysfunction caused by damage to the D1 protein (*Warner, 1999*) and host cells (*Nii & Muscatine, 1997*). The increase in the production of ROS is a stress signaling mechanism that can potentially trigger an oxidative stress and apoptotic cascade in coral cells (*Hensley et al., 2000*; *Drury et al., 2022*).

The relationship between macroalgae and corals under climate change conditions remains inconclusive. To investigate the effects of macroalgae on hermatypic coral under ocean warming, the massive coral, *Turbinaria peltata*, and macroalga, *Caulerpa taxifolia*, which are common species with frequent interactions in the Xuwen Sea, were selected as

study species. *C. taxifolia* is a multinucleate siphonous green alga and is known to have great invasive potential worldwide (*Zubia et al., 2020*). Furthermore, it has been found that *C. taxifolia* can produce potential allelochemicals, such as monoterpenes and sesquiterpenes (*Guerriero et al., 1992*, *1993*). Given that *C. taxifolia* usually grows on various hard substrates close to large numbers of live coral colonies, the physical and chemical impact has to be better understood. To evaluate the effect of its chemical and physical effects, an indirect contact group was used to investigate chemical effects, and the direct-contact group was used to explore the combined effects of physical and chemical processes. In this study, we show that increased temperature, representing the ocean-warming range projected for this century (*Spady et al., 2022*), enhances the ability of seaweeds to impact the physiology of corals, potentially shifting the competitive interaction between corals and seaweeds in favour of seaweeds. It provides a reliable basis for the evolution of competition between corals and macroalgae under future global changes.

## MATERIALS AND METHODS

Part of this text were previously published as part of a preprint (https://www.researchsquare.com/article/rs-1878684/v1).

### Sample collection

*T. peltata* (with a skeleton size of length:width:height = 20:18:12 cm, one colony) and *C. taxifolia* (5 kg) were collected from the Xuwen Coral Reef National Nature Reserve (109° 55′ E, 20° 16′ N) at a depth of approximately 4 m. The samples were transported to the laboratory and cultured in two 200-L tanks at a temperature of 26.5 °C, pH of 8.0, salinity of 33, and 200 μmol photons $m^{-2}s^{-1}$ with a 12: 12 h light/dark cycle for 3 months. Blue lighting was provided by a photosynthetically active radiation (PAR) lamp (Aqua Knight M029). An illuminometer (UNI-T UT383) was used for light measurement. After 3 months of acclimation, the corals from the same colony were cut into 54 pieces, approximately 4 cm in diameter, and were fixed on a ceramic base with aqua rubber (Aron Alpha GEL-10). The samples were then placed in another 200-L tank for 1 week until the anthopolyp had stretched naturally.

### Experimental design

After acclimation, 54 coral nubbins were randomly allocated into 10 L experimental tanks. To mimic the coral-macroalgal interaction in coral reefs, the same amount of macroalgae was co-cultured with the coral samples in the following ways: (1) no algae were added to the tank, *i.e.*, the control group (Fig. 1A). (2) Algae (25 g) were cultured in external algae boxes with no direct contact between the algae and the coral samples. The allelopathic substances produced by algae could enter the tank with the corals by water flow to determine whether the algae had a water-mediated interaction in the co-occurrence group (Fig. 1B). (3) The algae and corals were co-cultured in the same tank with direct contact, and a panel made of polymethyl methacrylate (PMMA) was used to fix the height of the algae parallel to the coral samples; this was referred to as the direct contact group (Fig. 1C).

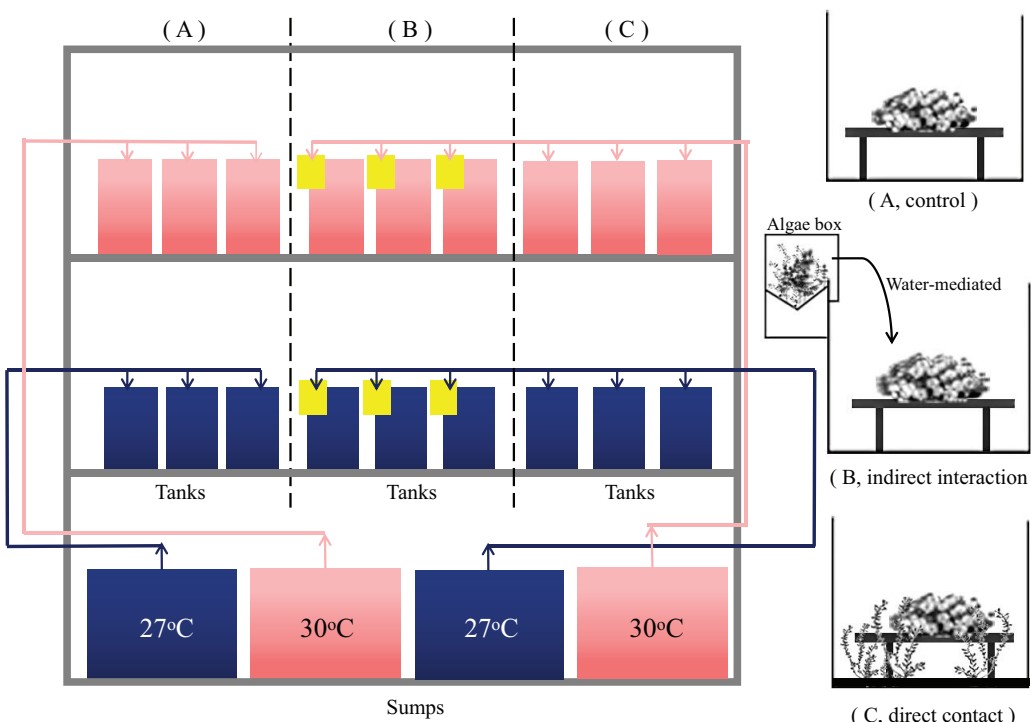

**Figure 1 Experimental operating system.** System control with a feedback loop to adjust the conditions. Seawater at 27 °C (blue) and 30 °C (red) with a feedback loop to adjust for the water temperature was heated in different collection sumps (36 L) and then fed into each tank (10 L). There is a drain at the bottom of each tank. The control group had a separate sump and the algae treatment group shared a sump. Macroalgae treatments were applied to (A) *T. peltata* (B) *T. peltata* indirect interaction with *C. taxifolia*. Water flowed first into the algae box (yellow) and then into the tanks to deliver the allelopathic substances secreted by the algae. (C) *T. peltata* in direct contact with *C. taxifolia*. Panel made of PMMA was used to fix the height of the algae parallel to the coral samples. Each treatment contained three replicate tanks, within which three coral nubbins were placed.

The experimental tanks were subjected to ambient conditions (27 °C) and the shared socioeconomic paths (SSPs) scenario SSP2-4.5 (30 °C) (*Zhongming, Linong & Xiaona, 2021*), to mimic the typical temperature range that the corals experienced at Xuwen Coral Reef Nature Reserve in order to explore the coral-macroalgae contacts at different temperatures. Each treatment contained three replicate tanks, within which three coral nubbins were placed per tank.

The temperature in each experimental group was increased to the set temperature by 1 °C per day. The first 3 days were the temperature adjustment period. Some algae tips decomposed due to metabolism or blue light intolerance. Therefore, the algae were checked and replaced each day to ensure that the experimental group had 25 g (0.0025 g cm$^{-3}$) of fresh algae, which is the amount of algae with the density of 0.0022 g per cubic meter of water surveyed from the inshore reef of Xuwen Coral Reef Nature Reserve. Fifty percent of the seawater was replaced every 3 days in each tank. Organisms were kept under treatment conditions for a period of 28 days and physiological measurements were subsequently performed.

### Endosymbiont density and Chl $a$ content

At the end of the experiment, coral tissue were removed from the nubbins using a waterpick (0.45 μm filtered seawater), and the slurry was homogenized. Six 15 ml samples were taken from the slurry, centrifuged (4,000 rpm min$^{-1}$, 4 °C, 10 min), and the supernatant was removed. Part of the pellet was suspended in 5 mL formaldehyde to count the endosymbiont density under an inverted microscope (DMI 6001B, magnification eyepiece × magnification objective: 10 × 20) with a blood counting plate (CKSLAB CB30). Another portion was resuspended in 8 mL methanol. The pigments were extracted at 4 °C for 24 h. The extract was centrifuged (4,000 rpm min$^{-1}$, 4 °C, 10 min), and Chl $a$ was determined according to the method described by *Ritchie (2006)*. Data were normalized to skeletal surface with the aluminium foil technique (*Marsh, 1970*).

### Feeding rate

A total of 1 ml of *Artemia* solution was added to 1 L of water and a plankton counter was used to measure the mixed solution and to determine the individuals of the *Artemia* solution per milliliter. Nubbins were moved into the feeding tanks (1 L) with an *Artemia* concentration of ~ 2 ind mL$^{-1}$, while one tank served as a control (without coral). After an incubation period of 1 h, the coral nubbins were rinsed with seawater and returned to their respective positions in the experimental tanks. The feeding rate was calculated as the decline in *Artemia* concentration in the feeding tanks and normalized per polyp. The measurement was performed once a week between 11:00–12:00 a.m. The number of polyps in each nubbins were visually counted before the experiment.

### Growth rate

The coral nubbins were weighed on a balance (UW 2200H, accuracy = 0.01 g) using the buoyant weight technique (*Davies, 1989*). Before each measurement, the surface of the coral ceramic base was lightly brushed with a toothbrush to remove algae. A glass beaker was filled with 1 L of filtered seawater (27 °C, salinity 32). Then the coral nubbins were placed on the bottom of the beaker and the weight (minus the weight of the beaker without samples) was measured. The growth rate (g d$^{-1}$) was calculated as $(M_{ti} - M_{t0}/T_i)$, where $M_{t0}$ represents the nubbin weight at the beginning of experiment, $M_{t1}$ the weight at the measureing time point and $T_i$ represents the duration in days. The measurement was repeated every 7 d. Data were normalized to skeletal surface area determined with the aluminium foil technique (*Marsh, 1970*).

### SOD and CAT

The homogenized coral tissue slurry that was used to measure the endosymbiont density and Chl $a$ content was centrifuged (4,000 rpm min$^{-1}$, 10 min, 4 °C), and the supernatant was collected to measure the SOD and CAT activities. These measurements were determined in the dilution using kits (A001-1-1, A007-1-1; Nanjing Jicheng, Nanjing, China). A BCA (Bicinchoninic Acid Assay) kit was used to determine the protein concentration (A045-3-1; Nanjing Jicheng, Nanjing, China). The enzyme activities were normalized to total protein content as U mg prot$^{-1}$.

## Data analysis

The results are presented as the means ± standard deviations. Data were tested for homogeneity of variance (visual inspection of residuals *vs.* fitted values), and the normality of the residuals was tested using the Shapiro–Wilk normality test. All response data of corals were tested using a two-factor analysis of variance (ANOVA) with "temperature (27 °C, 30 °C)" and "algae (control, direct contact, indirect interaction)" as fixed factors, including the interaction term. When significant effects of factors occurred, ANOVAs were followed by Tukey's multiple comparisons test for identifying differences between algal treatments at the same temperature and Sidak's multiple comparisons test for identifying differences between temperature treatments. The data was analysed and graphed using GraphPad Prism 8.0 and $p < 0.05$ was considered a significant difference (the statistical results of $p$-value supported by the Supplemental Material).

# RESULTS

## Seawater chemistry monitoring

The temperature, pH, and salinity values were measured continuously during the experiment (Fig. 2).

## Endosymbiont density and Chl *a* content

Figures 3A and 3B show that the density and pigment content of the endosymbiont were significantly influenced by temperature. Without algae, elevated temperature resulted in an average 44.6% (from 1.3 ± 0.1 to 0.72 ± 0.07 × $10^6$ cells cm$^{-2}$) decrease in endosymbiont density ($p < 0.01$) and a average 58% (from 17.1 ± 1.5 to 7.2 ± 2.6 µg cm$^{-2}$) decrease in Chl *a* content ($p < 0.01$). Compared with the control group, the co-occurrence group (0.95 ± 0.1 × $10^6$ cells cm$^{-2}$, $p = 0.03$) and direct contact group (0.89 ± 0.2 × $10^6$ cells cm$^{-2}$, $p = 0.01$) in 27 °C temperature treatments showed lower mean values of endosymbiont density. However, there were not significantly differences between the control, co-occurrence and direct contact groups at 30 °C. The Chl *a* concentration only differed significantly between the control and the direct contact group at ambient temperature ($p = 0.03$).

## Feeding rate

The results of the feeding rate are displayed in Fig. 4A. Primarily, elevated temperature had a significantly detrimental effect on the feeding rate among the treatments. There was significant difference between the the co-occurrence group ($p = 0.02$) and the direct contact group ($p = 0.03$) compared with control group at 27 °C. At 30°C, direct contact with algae caused the feeding rate to fall to a minimum of 12.8 ± 1.1 ind polyp$^{-1}$ h$^{-1}$ compared with control group ($p < 0.01$). The interaction of temperature and algae influenced the feeding rate significantly (F = 4.7, $p = 0.04$, Table 1).

## Protein content

The results of protein content are shown in Fig. 4B. At 27 °C, algae caused a loss of 37.5% of protein content in coral tissue in the co-occurrence group ($p = 0.01$) and 49.0% protein

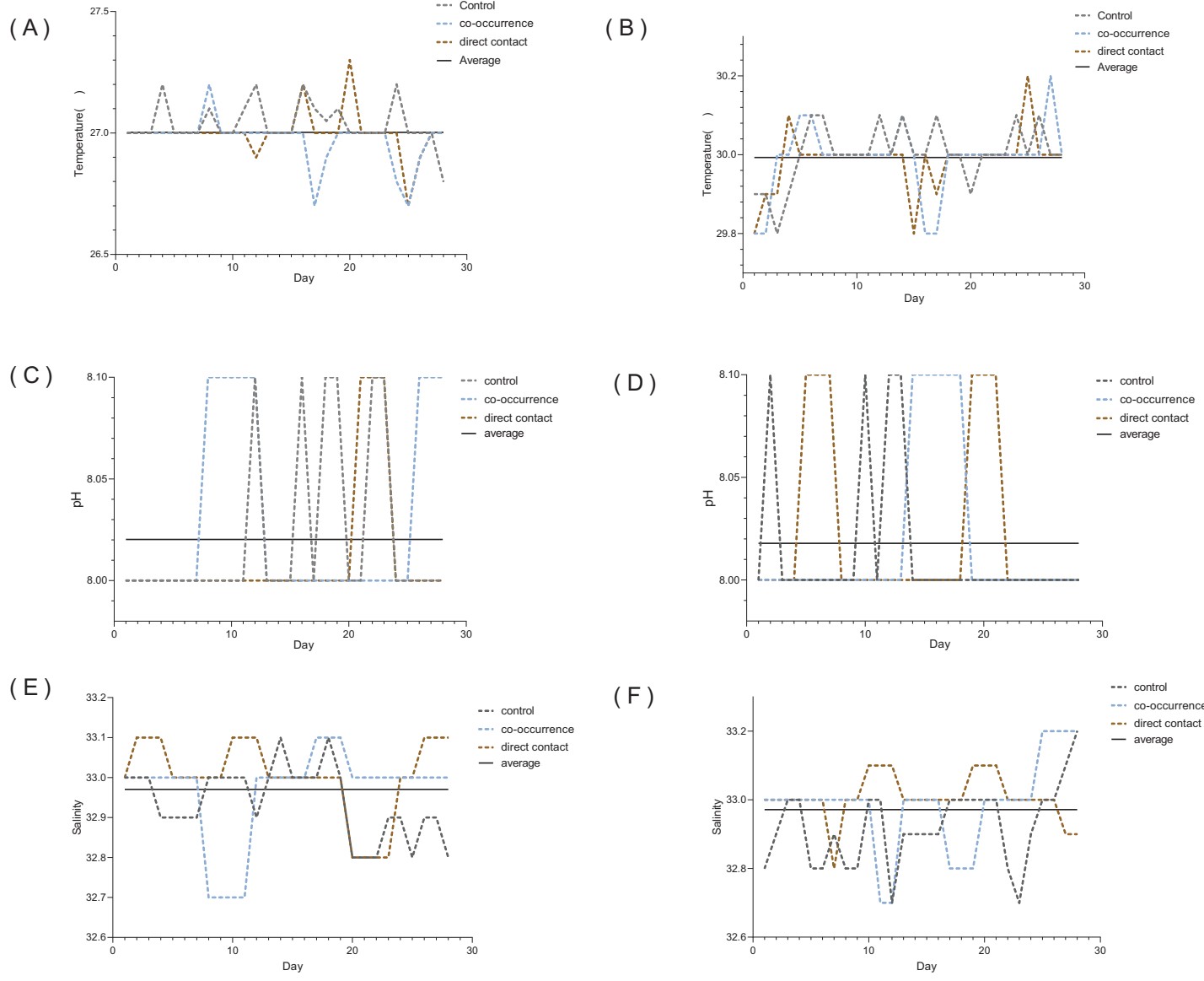

**Figure 2 The variability in the temperature, pH, and salinity over the course of the 4 weeks experiment.** Seawater at 27 °C (A) temperature, (C) pH, (E) salinity. Seawater at 30 °C (B) temperature, (D) pH, (F) salinity. The dotted lines represent control, direct contact and co-occurrence groups. The solid line represents the average.

content in the direct contact group ($p < 0.01$). Although contact with algae further decreased the mean of protein content, the difference was not significant with the co-occurrence group. At 30 °C, the direct contact with algae resulted in the lowest protein content of $1.2 \pm 0.3$ mg cm$^{-2}$ ($p < 0.01$, $p < 0.01$) compared with the control and co-occurrence groups. No significant difference was observed between the temperature treatments in the presence or absence of algae ($p = 0.11$, Table 1). There was no significant interaction between algae and the temperature treatment (F = 2.86, $p = 0.12$, Table 1).

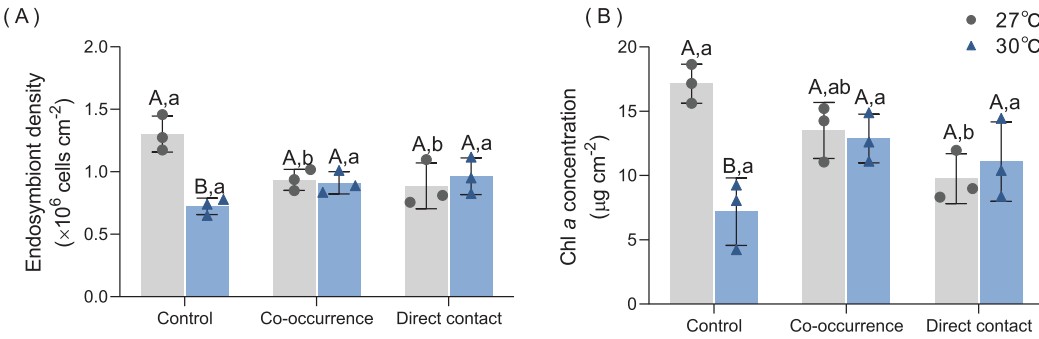

**Figure 3** The effect of 27 °C (grey) and 30 °C (blue) temperature treatment on the (A) endosymbiont denstiy and (B) Chl *a* concentration of corals in direct contact or co-occcurance with macroalgae. Uppercase letters represent differences between temperature treatments, lowercase letters represent differences between algal treatments at the same temperature, and different letters represent significant differences (*p < 0.05*). Data are expressed in terms of the mean ± standard deviation, *n* = 3.

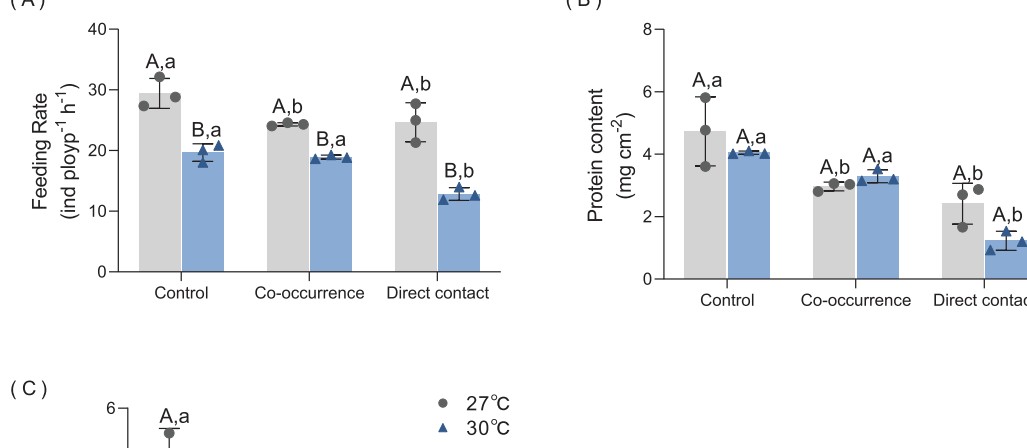

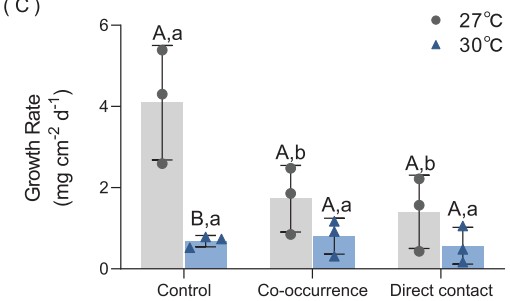

**Figure 4** The effect of 27 °C (grey) and 30 °C (blue) temperature treatment on the (A) feeding rate, (B) protein content and (C) growth rate of corals in direct contact or co-occcurance with macroalgae. Uppercase letters represent differences between temperature treatments, lowercase letters represent differences between algal treatments at the same temperature, and different letters represent significant differences (*p < 0.05*). Data are expressed in terms of the mean ± standard deviation, *n* = 3.

## Growth rate

As indicated by the change in buoyant weight, the growth rate was affected by algae at 27 °C and the effect of elevated temperature was only seen in the control group (Fig. 4C). At 27 °C, the growth rate of corals in the control group was highest, with a mean value of

**Table 1 Two-way ANOVA output of different variables for *T. peltata*.**

| Variable | Source of variation | F | *p* |
|---|---|---|---|
| Growth rate | Algae | F(2,8) = 28.80 | **<0.01** |
| | Temperature | F(1,4) = 7.78 | **0.049** |
| | Interaction | F(2,8) = 28.00 | **<0.01** |
| Feeding rate | Algae | F(2,8) = 14.57 | **<0.01** |
| | Temperature | F(1,4) = 119.80 | **<0.01** |
| | Interaction | F(2,8) = 4.70 | **0.04** |
| Endosymbiont density | Algae | F(2,8) = 0.75 | 0.5 |
| | Temperature | F(1,4) = 21.94 | **0.01** |
| | Interaction | F(2,8) = 9.05 | **0.01** |
| Chl *a* | Algae | F(2,8) = 1.58 | 0.27 |
| | Temperature | F(1,4) = 90.92 | **<0.01** |
| | Interaction | F(2,8) = 7.29 | **0.02** |
| Protein | Algae | F(2,8) = 31.87 | **<0.01** |
| | Temperature | F(1,4) = 4.28 | 0.11 |
| | Interaction | F(2,8) = 2.86 | 0.12 |
| SOD | Algae | F(2,8) = 38.81 | **<0.01** |
| | Temperature | F(1,4) = 16.01 | **0.02** |
| | Interaction | F(2,8) = 2.37 | 0.16 |
| CAT | Algae | F(2,8) = 10.01 | **<0.01** |
| | Temperature | F(1,4) = 64.48 | **<0.01** |
| | Interaction | F(2,8) = 5.13 | **0.04** |

**Note:**
The bold values indicate the significant effects on the variable. F = F value; *p* = *p* value (significant < 0.05).

4.1 ± 1.4 mg cm$^{-2}$ d$^{-1}$. Compared with control group, the coculture with macroalgae decreased the growth rate of coral by 57.7% in the co-occurrence group ($p < 0.01$) and 65.5% in the direct contact group ($p < 0.01$). But no significant difference between co-occurrence and direct contact group. Elevated temperature had an inhibitory effect on the growth rate in the coral culture system without algae. The growth rate in this group was 83.4% lower than the control group ($p < 0.01$). The elevated temperature combined with direct contact with algae resulted in the lowest coral growth rate, with a value of 0.57 ± 0.45 mg cm$^{-2}$ d$^{-1}$. However, the differences among algae treatments at an elevated temperature were not significant. The interaction of temperature and algae influenced the growth rate significantly (F = 28, *p < 0.01*, Table 1).

## SOD and CAT

As shown in Fig. 5A, macroalgae treatments increased the SOD antioxidant capacity of corals under both temperature conditions. At 27 °C, co-occurrence with algae increased the SOD activity of coral 1.85-fold compared with control group (*p = 0.03*). Moreover, the SOD activity was higher for the direct contact group (288.1 ± 16.6 U mgprot$^{-1}$) than control groups ($p < 0.01$) and co-occurrence group (*p = 0.03*). At 30 °C, the mean SOD activity of coral without the presence of algae increased by 1.77-fold compared with its

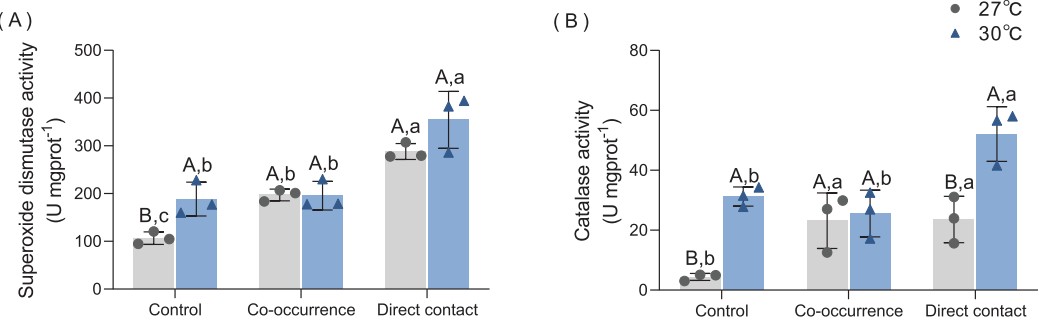

**Figure 5** The effect of 27 °C (grey) and 30 °C (blue) temperature treatment on the (A) superoxide dismutase activity, (B) catalase activity of corals in direct contact orco-occcurance with macroalgae. Uppercase letters represent differences between temperature treatments, lowercase letters represent differences between algal treatments at the same temperature, and different letters represent significant differences ($p < 0.05$). Data are expressed in terms of the mean ± standard deviation, $n = 3$.

counterpart at ambient temperature ($p = 0.03$). In the direct contact group at 30 °C, SOD activity increased to the highest level of 354.3 ± 59.56 U mg prot$^{-1}$ compared with control group ($p < 0.01$). However, in the co-occurrence and direct contact system, there was no significant difference caused by elevated temperature, indicating that both factors did not interact (F = 2.37, $p = 0.16$, Table 1).

The CAT activity also increased after algae interaction and elevated temperature, as shown in Fig. 5B. At 27 °C, co-occurrence with algae caused the CAT activity in coral tissue to rise 5.3-fold ($p = 0.046$), which is comparable to the level of CAT activity in the direct contact group ($p = 0.04$). At 30 °C, the CAT activity in the control also increased by 7.1-fold ($p < 0.01$). Moreover, when cultured in contact with the algae, the CAT activity further doubled compared with the control group ($p = 0.03$). The combined effect of temperature and macroalgae was significant (F = 5.13, $p = 0.04$, Table 1).

## DISCUSSION

This study explored the crucial issue of how the physiology and oxidative stress response of a hermatypic coral are affected by macroalgae at elevated temperatures. We set up three treatments of the macroalga *C. taxifolia* (direct contact, indirect, water-mediated presence, no alga) to act on the coral *T. peltata* at ambient temperatures (27 °C) and elevated temperature (30 °C). The results demonstrated that macroalgal presence increased the antioxidant activity at 27 °C. In addition, combined with elevated temperature, there was a remarkably synergistic effect that macroalgae impacted the feeding rate, protein contain and further increased the oxidative stress of the coral, in which contact with algae had a more severe effect than indirect interaction.

### Effects of *C. taxifolia* on endosymbiont of *T. peltata*

Algae was found to influence the average endosymbiont density and chl *a* content of *T. peltata*, however, no bleaching occurred. A number of studies have reported that coral's photosynthetic efficiency decreased (Fv/Fm), or bleaching occurred, when there was direct or indirect contact with macroalgae. However, not all coral species are equally susceptible

to algae and not all algae will have deleterious effects on corals (*Smith et al., 2006*; *Rasher & Hay, 2010*; *Fong et al., 2020*). *Rasher & Hay (2010)* suggested that *Padina perindusiata* and *Sargassum* sp. did not inhibit photosynthetic efficiency or induce bleaching of *Porites porites*, which might be explained by the fact that the 20-day interaction period was too short to impact the coral health. Additionally, *T. peltata* is a massive coral that could resist environmental pressure by increasing its basic metabolism (*Loya et al., 2001*). This may explain why there was no significant bleaching effects of macroalgae on endosymbiont density. There was a stronger effect for direct contact compared to co-occurrence in chl *a* concentration at 27 °C, suggesting that physical mechanisms are still the main way in which macroalgae affect corals.

### Effects of *C. taxifolia* and thermal stress on the physiology of *T. peltata*

The feeding rate was affected by elevated temperature. *Johannes & Tepley (1974)* also found that the feeding rate of coral decreased in heat stress because of the polyp contraction or a loss of nematocyst function. Our results suggest that a decrease in chl *a* content and endosymbiont density was the reason why the feeding rate was impacted in elevated temperature. Endosymbionts provide photosynthate to host cells (*Van Oppen & Blackall, 2019*). A decrease in the endosymbiont density at high temperatures may result in reduced energy expenditure to maintain normal physiological functions and reduce resistance to predation. This study showed that contact with *C. taxifolia* resulted in the greatest reduction in the feeding rate at 30 °C. In summary, thermal stress might a crucial factor affecting the feeding ratio of *T. peltata*, which became more severe when in contact with macroalgae.

Macroalgae can induce reduced protein content in corals. Damage to coral tissue by contact with macroalgae has been documented in many studies. *Bender, Diaz-Pulido & Dove (2012)* asserted that *Acropora* sp. lost tissue and decreased its growth rate due to allelopathy mechanisms after coming into contact with *Chlorodesmis fastigiata*. In fact, macroalgae may transfer many allelopathic substances to corals, altering the structure of the microbial community and impacting the physiological processes of corals (*Fong et al., 2020*). This damage may ultimately reduce the protein content. Under stress, massive corals with thicker tissues may overcome the effects of endosymbiont loss through catabolism (*DeCarlo & Harrison, 2019*). Macroalgae may affect coral tissue by creating anoxic zones. *Barott et al. (2009)* demonstrated that after the interactions between corals (*Pocillopora verrucosa*, *Montipora* sp.) and some species of macroalgae (*e.g.*, *Gracilaria* sp., *Bryopsis* sp., and various turf algae), the characteristic patterning of coral pigments and polyps was altered and the tissue appeared damaged.

The growth rate of coral was altered by the macroalgae and temperature, the results were consistent with previous studies (*Tanner, 1995*; *Rölfer et al., 2021*; *Vega Thurber et al., 2012*; *Vermeij et al., 2009*). At 27 °C, the growth rate of the direct contact group was lower than that of the co-occurrence group. Therefore, the effects of seaweed on coral growth may require direct contact at ambient temperatures (*Clements et al., 2020*). *Brown et al. (2019)* also demonstrated that coral growth was reduced or even negative at 30 °C when in contact with algae. *Longo & Hay (2015)* determined that corals showed signs of stress at

30 °C, and contact with *Halimeda heteromorpha* further contributed to a decreased growth rate and increased mortality rate. These results may be due to the simultaneous decline in the autotrophic and heterotrophic activities of corals under the impacts of thermal stress or macroalgae, resulting in a drop of protein content and ultimately affecting the growth rate.

### Effects of *C. taxifolia* and thermal stress on oxidative stress of *T. peltata*

Corals under thermal stress may produce ROS (*Blanckaert et al., 2021*). *Downs et al. (2002)* documented that when exploring the varied oxidative stress response of coral under seasonal change, the SOD in summer was three times higher than that in winter. This study determined that SOD in corals was higher when macroalgae were present (the effect was stronger for direct contact compared to co-occurrence group), the increased temperature, or there was the synergistic effect of both. Thus, weakened corals were found to be more vulnerable to competition from algae, which was also supported by the results of *Diaz-Pulido et al. (2010)*. The level of both antioxidant enzyme activities was similar to thermal stress alone when *C. taxifolia* indirectly contacted *T. peltata*. These results indicate that the stress triggered by macroalgal allelochemicals on coral was equivalent to that induced by increased temperature. The temperature effect was stronger in CAT activity compared with SOD activity under direct contact, which may be related to the reduced protein content in coral tissues caused by elevated temperature under the direct contact treatment. Due to the evident decrease in the protein content of coral tissues in the direct contact group, the amount of antioxidant enzymes produced by coral is not enough to resist the damage of ROS.

## CONCLUSIONS

The shift from coral dominance to algal dominance that has been observed in many reefs due to global climate change and overfishing. Coral dominance is sensitive to key algal groups and other benthic groups, and shifts in ecosystem phases have a noticeable impact (*Tebbett et al., 2023*). The results of this study showed that *C. taxifolia* negatively affected the endosymbiont density, chl *a* content, feeding rate, growth rate and protein content of *T. peltata* and increased the antioxidant activity at 27 °C. The combination of elevated temperature and macroalgae interactions may further exacerbate the adverse effects on corals. Future studies are needed to explore the interactions of multiple coral-macroalgal species under climate change. Because of the vulnerability and sensitivity of coral reef ecosystem, relevant entities should take urgent steps to prevent $CO_2$ emissions that exceed the goals of the Paris Climate Agreement. Herbivorous fish populations should also be restored to improve macroalgae management in reefs.

### Funding

This work was financially supported by the National Key R&D Project of China (Grant No. 2022YFD2401302), GuangDong Basic and Applied Basic Research Foundation (Grant

No. 2019A1515110225), Guangdong Innovation and Strengthening School Project (Grant No. 230419080), and the Program for Scientific Research Start-up Funds of Guangdong Ocean University (Grant No. R18024). The funders had no role in study design, data collection and analysis, decision to publish, or preparation of the manuscript.

## Grant Disclosures

The following grant information was disclosed by the authors:
National Key R&D Project of China: 2022YFD2401302.
Basic and Applied Basic Research Foundation: 2019A1515110225.
Innovation and Strengthening School Project: 230419080.
Scientific Research Start-up Funds of Guangdong Ocean University: R18024.

## Competing Interests

The authors declare that they have no competing interests.

## Author Contributions

- JianRong Fu conceived and designed the experiments, performed the experiments, analyzed the data, prepared figures and/or tables, authored or reviewed drafts of the article, and approved the final draft.
- Jie Zhou conceived and designed the experiments, performed the experiments, analyzed the data, prepared figures and/or tables, authored or reviewed drafts of the article, and approved the final draft.
- JiaLi Zhou conceived and designed the experiments, performed the experiments, authored or reviewed drafts of the article, and approved the final draft.
- YanPing Zhang conceived and designed the experiments, performed the experiments, authored or reviewed drafts of the article, and approved the final draft.
- Li Liu conceived and designed the experiments, performed the experiments, authored or reviewed drafts of the article, and approved the final draft.

## Data Availability

   The raw measurements are available in the Supplemental File.

## Supplemental Information

Supplemental information for this article can be found online at http://dx.doi.org/10.7717/peerj.16646#supplemental-information.

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
