# Peer review of "Competitive effects of the macroalga *Caulerpa taxifolia* on key physiological processes in the scleractinian coral *Turbinaria peltata* under thermal stress"

_PeerJ, doi:10.7717/peerj.16646_

## Round 0.1 · original submission · Major Revisions

The paper addressed new interesting findings about the interaction between Caulerpa taxifolia and Turbinaria peltata. Nevertheless, some rewriting is needed and the reviewers have given you several comments that you should take into account in the new version.

·

Basic reporting

The basic formal requirements (i.e. article structure, no discussion in the result part, etc) in this article are met. However, several issues exist. First of all, there are many grammatical and other language mistakes, which make some sections difficult to understand (e.g. lines 105 - 107). Some (but not all) of these issues are annotated in the PDF I uploaded, but there are many more issues. I recommend that a professional editor checks the manuscript or that a co-author with good knowledge of English is added.
In the article, three experimental treatments are compared: Corals in direct contact with algae, corals and algae that are not in direct contact, and corals without algae. In the article, this is often described as interactions. This implies that the coral and the alga react to the presence of the other organism. This is not necessarily the case. Therefore, I recommend renaming the treatment groups, e.g. into "direct contact", "co-occurrence" (or something similar), and "control". The names should be used consistently in the article.
The introduction and cited literature give a sufficient background into the topic and the aim of the study is described. The specific research questions or hypotheses are not described and could be added. The impact of herbivorous fishes is mostly ignored, a small section could mention that overfishing and nutrient pollution is the main reason for phase shifts towards macroalgae (e.g. Barott KL, Williams GJ, Vermeij MJA, Harris J, Smith JE, Rohwer FL, Sandin SA (2012) Natural history of coral−algae competition across a gradient of human activity in the Line Islands. Mar Ecol Prog Ser 460:1-12. https://doi.org/10.3354/meps09874).
The figures and tables are in general appropriate, however, some issues remain in the descriptions.
Figure 1 (regarding the aquaria set-up) is not clear and should be improved, for me, it is unclear if all aquaria of one temperature group are fed by one water supply system or not. In the other figures, it is not clear, why sometimes, letters indicate significantly different groups and why they are sometimes missing.

Experimental design

The experimental setup could give valuable information that so far, especially for this region, does not exist.
No research questions or hypotheses are described. Please add this to the manuscript.
The Materials and Methods section has to be improved. Often, details are not described sufficiently and parts are missing (see annotated PDF for details). It is for example not clear, how the pellet mentioned in lines 153 - 155 is divided for chl a and symbiont density measurements and on which part the enzyme activity measurements described in part 3.6 were performed. Also, the exact aquaria setup is not clear. Parts visible in Figure 1 are not described in the text. It is e.g. also not described, how the algae were attached to the coral fragments and what an "algae box" might be and how the suspected allelochemicals might contaminate other tanks. Additionally, the quality of English makes it sometimes hard to understand some parts
The statistical analysis could be improved, too. As far as I understood, for each parameter, the mean of three coral (pseudo)replicates was calculated before further analysis. All data could be analysed with a mixed effects model including a random factor for the aquaria ID, and of cause, the calculation of the mean has to be described.
An ANOVA was used to analyse the data consisting of the interaction of the temperature treatment group (27°C and 30°C) and the alga group (direct contact, "co-occurrence", control). A posthoc test was performed to understand the differences between groups in more detail, but it is not clear why an LSD test was used for the algal treatments and a Tukey test for the temperature treatment (or I misunderstood the analysis).

Validity of the findings

The raw data is shared but I think, either data is missing or it is not described that means were calculated during the analysis. In the Material and Methods parts is described that each experimental group consisted of 3 aquaria with 3 coral nubbins, i.e. 9 data points should exist per group. But in the plots and in the shared data, there are only 3 values per group.

The reporting of statistical results has to be improved. The section is very confusing, it is often not clear which groups are compared. Describing the outcome of interactions (in the ANOVA) is difficult but it is possible to make the comparisons and statistical results more clear. Often, F-values are missing and sometimes, two p-values are included and it is not clear which value belongs to which comparison.

In Figure 4C it is unclear, why there should be significant differences in the 27°C treatment group between the growth rates of corals in direct contact with algae and in indirect contact. The values are basically the same but the letters suggest significant differences. In Table 2, horizontal lines could help to directly understand which value belongs to which group.

One main outcome, also mentioned in the abstract should be removed from the manuscript. The experiment cannot be used to state that algal contact has the same effect on corals as ocean warming.

The authors mention high variability in the pH, salinity, and temperature over the course of the experiment. However, just a table with mean values and standard deviations is given. Since these parameters were continuously measured, a plot would be useful to get an understanding of the variability. Also, this data is missing from the shared raw data.

Additional comments

In my opinion, this will be a very nice article adding interesting findings for a region for which few data exist. Clearly, the article has to be improved but the authors performed an interesting experiment and analysis and I hope that it will be published soon. Since for now, the manuscript does not yet fulfil the quality standards for a scientific publication, I recommend major revisions. I am happy to review a revised version of the manuscript.
Since the organisms were collected in a nature reserve, permits might be needed. Whether that is the case is not mentioned in the article.

Reviewer 2 ·

Basic reporting

no comments

Experimental design

The experimental design is clearly explained and the authors provide all the necessary technical details to replicate what they have done.

Validity of the findings

The manuscript's findings are well described and have a significant impact on the scientific community.

Annotated reviews are not available for download in order to protect the identity of reviewers who chose to remain anonymous.

---

## Round 0.2 · Major Revisions

I apologize for the delay in the decision; the previous editor is apparently busy and I have stepped in to guide your submission. I have looked over the one review received this round (a re-review), and read your work myself. I agree with the reviewer that although your paper has improved from the first submission, much work remains, and your improvement remains limited. Please consider the reviewer's comments very carefully; in the interest of moving your work forward please note I may reject your work without substantial improvement and consideration of the reviewer comments.
As well, on my own reading of your work, I note:

1. species name spelling mistakes and such things, which indicate to me a lack of attention to detail. "sp." should not be in italics.

2. Please explicitly note you used n=1 colony (line 126 - is this correct?).

3. Please use the scientific name of "endosymbiont" at least once, upon first mention.

I look forward to seeing an improved version.

·

Basic reporting

The article improved since the last version. However, there are still many issues regarding the language, grammar, and figures. There are still many typos, even in the title (scleratinian instead of scleractinian), species names (Turbinaria peltate instead of Turbinaria peltata, for many times in the manuscript), and the response to the last review.

I indicated some issues (but not all) in the PDF and compiled below a list of the most important mistakes:

Overall, interaction still used where rather contact should be used.

Line 57: microalgae: You mean macroalgae

Line 62: “of” should be “and”.

Line 57: “may affect the physiological responses of corals in different ways”. This is very ambiguous. Describe clearer what you did, i.e., investigate the response to competition.

Line 59: “photosynthesis performance of corals”. No, of the Symbiodiniaceae

Line 65: “oxidative stress” should be “oxidative stress response”.

Line 71: What is “tissue recovery”?

Line 74: “the oxidative stress of Acropora millepora was activated” “response” missing.

Lines 72 - 76: Here is a lot of information missing. What do you mean by damage to gene structure and altered transcription factors? Explain briefly!

Lines 75 - 76: Instead of “…after coming into contact with the macroalga Chlorodesmis fastigiata thalli and their hydrophobic extract over a short-term period” , rather write: “..after contact with the macroalga Chlorodesmis fastigiata and its hydrophobic extract…

Lines 79 - 80 :What is “light calcification rate”?

Lines 80 - 81: The authors still confuse/mix “contact” and “interaction”. Here both terms were used: “by the interaction of Chlorodesmis fastigiata contact”

Line 93: replace “fluctuate” with e.g. “which are impacted by”

Line 95: Which years were taken into account for the mean?

Lines 96 - 98: “When compared with the DHW in the coral reef, it is plausible that the physiological responses of corals in the Xuwen Coral Reef National Nature Reserve of China may be affected by thermal stress (Spady et al., 2022).”: Not clear. Which coral reef are you referring to? What is “the DHW in the coral reef”? What time? Which reef? Which values of DHW?

Line 99: What is decomposing tissue proteins?

Line 101: “ROS, a substance that damages cells by accumulation” Very vague. And it was explained further up. (Line 72, improve the explanation there)

Line 103: “or” should be “of?”

Line 154: what do you mean with faster life cycle? I think the algae were stressed and died?

Line 172: invert? Wrong word. Determine?

Line 173: When introduce an abbreviation (ind), explain what it stands for (individuals).

Lines 182-184: unclear: See suggestions in pdf

Lines 192: BCA not explained

Line 201: “diûerences”: There seems to a rendering issue in the pdf.

Line 185: parameter M_t1 not explained

Line 206: “The standard deviation is due to daily variations.” No. Either the high value of the SD or the variability.

Line 209: Rather use elevated temperature than “thermal stress”

Lines 215 - 218: This sentence is an interpretation of the results and should therefore not be included in the results but in the discussion. Additionally, I think you cannot do this assumption. The only significant difference between control and direct contact group was at 27 °C and only for Chl a. So no, I think there’s no “antagonistic” effect of temperature and algal treatment

Lines 223 - 224: Similar, synergistic is an interpretation that should be made in the Discussion. I also think that you cannot make this conclusion with your results!

Line 230: What do you mean with a “non obvious” difference? Non significant?

Line 258. Missing point after sentence

260 “As the temperature increased”. Don’t not use this, you did not investigate the impact of increasing temperature (implying a process) but rather between two temperatures. This expression was used a couple of times, change it to the exact treatments

Line 295: “the predation of T. peltate”. No, this implies that it was predated. Use feeding rate as before. Additionally, peltata instead of peltate

Lines 312 -213: “corals weakened at 30°C” . Corals showed signs of stress? What do you mean by weakened?

The figures and figure descriptions need to be improved:

Figure 1:
Title: “and then fed into the each tanks (10 L).” . Change e.g. to: “And then fed into each tank”.

To make the plot more understandable, instead of A, B, C, use control, indirect interaction, and direct contact.

Put the side panel above the plot so it corresponds with the columns of the tanks below.

What are the grey lines and boxes? Is it just to control the temperature? I’d leave this away. Just show how he water flows. Indicate this with arrows.

What about the outflow from each tank? Where did it go?

What is the blue gradient above each tank? Does it symbolise a lamp? This could be left away. Focus on the water flow

Why mention polymethyl methacrylate? If I understood correctly, it is the material that was used for some type of fixation? The material of this is not important for the figure.

The text on dark blue background not readable. Use e.g. white text colour

What are the arrows for A and C in the right panel?

Figure 2:
Title: The dotted lines represent deviation values for direct and co-occurrence groups of environmental factors. What are deviation values?

Why no dotted lines in A?

Maybe add captions above A, C and E (no algae) and B, D, and F (with algae).

Why is there no plot for direct contact and indirect interaction?

I’d include one panel for each group (control, direct contact, indirect interaction), show the mean as a thick line and the individual tanks as a dotted line.

A/B: Why is the area below the line filled here and not for the other plots? Don’t fill the area.

C-F: zoom in the y axis to better see differences. Variability is much less than scale of y axis, so the variability is not visible

Figure 3 - 5:
Upper case letters. These are differences between temperatures at same treatment???

“corals treated by macroalgae after 4 weeks”: Rather corals in direct contact or co-occcurance with macro algae

Figure 3:
Title: Endosymbiont and chlorophyll a: add the word “concentration”, don’t use abbreviations (chl a) in figure titles.

Figure 4:
B) Add the word protein content

Figure 5:
Don’t use abbreviations (SOD, CAT).

Experimental design

As stated before, the data and experiment is very interesting and should be made available to others. However, some issues remain even in the updated version.

Line 126: “with a skeleton size of length:width:height = 20:18:12.” What are the units

Sample collection: How many colonies were collected?

Line 133: What is “with aqua rubber”? Which company, material etc

Lines 133 - 134: The samples were then placed in another 200-L tank for one week until the tentacles grew. Explain what do you mean with “until the tentacles grew”?

Lines 143 - 144: “and polymethyl methacrylate fixation was used to fix the height of the algae parallel to the coral samples.” What was used for the fixation? Polymethyl methacrylate is just a type of plastic? What day you mean?

Line 150: You refer to 4 algae treatments, but in the explanation (lines 138 - 145) only 3 are mentioned (and done when I understood correctly). They also mention 6 overall treatments (2 temp treat per algal treat) so I assume 3 is meant.

Line 200: In the response, you mentioned that you used a mixed effects model with tank ID as random factor. Explain this in the text! Random slope or random intercept? Which family was used? Gaussian?
Why did you sometimes used a LSD and sometime an Tukey’s test?

Validity of the findings

Additional to my last comments, I want to add the following:

It is interesting, that that the temperature only plays a role for the control group (symbionts, chl a, growth rate) and that there was no impact of temperature for protein content. Do you have an explanation?

In general, there was a stronger effect for direct contact compared to co-occurance (if there are differences, usually between control and direct contact, as for feeding at 30°C, growth rate at 27 °C, chl a at 27 °C, SOD). Maybe make this clearer in the discussion.

You see a consistent impact of temperature within the same treatment for feeding rates.

For Chl a, you see a significant difference at 27°C but to at 30 °C. At 27°C, the chl a content is sig. higher for the control than for the direct contact treatment, whereas the co-occurence treatment did not significantly differ to control and direct contact. No sig. differences between algae treatments at 30°C! Do you have an explanation for that?

Lines 211 - 213: “Compared with the control group at ambient temperature, corals that interacted with macroalgae in both temperature treatments showed lower mean values of endosymbiont density and chl a content.”
But this is not significant, right? In the Figure 3A, the lowercase letters are all “a”. So not only no significant difference between the two algae treatments (As you write in the next sentence) but also not to the control! But not sure if I understood correctly.

236: As indicated by the change in buoyant weight, the growth rate (Fig. 4C) was affected by both algae and thermal stress
Only for control!

Line 238 . 240: Compared with control group, the coculture with macroalgae decreased the growth rate of coral by 57.7% in the co-occurrence group (p = 0.06) and 65.5% in the direct contact group (p = 0.03).

But no sig difference between control and cooccurrence and co-ocurance and direct contact (looking at the letters in the plots)

254: Also no impact of temperature for direct contact!

---

## Round 0.3 · Minor Revisions

Dear Author,
the reviewer and I think you have improved the manuscript that now needs only minor changes. Please, fulfill these minor suggestions and send back to us the revised manuscript.

·

Basic reporting

The article has improved and is ready for publishing after editing some language errors (see below and in the annotated PDF) and a clarification of the statistical test (multiple comparisons in LSD test); see below.

Line 49: Global double, remove “at the global scale”
Line 93: Remove “which”
Lines 96f: “The DHW is an accumulation […]” This sentence could be removed, DHWs are not mentioned anymore in the introduction
Line 97: “When compared with the 2021-2022 DHW from […]”: No! You’re not looking at DHWs here but SST. Additionally, What are you comparing? 2020 vs 2021/2022? When you compare, state the difference, eg. “[…] the average SST was 1 °C higher.”.
Line 98: “was” instead of “is”
Line 101: What do you mean by “depleting protein”? I’d skip “for depleting protein” to make the sentence more clear.
Line 117: Add “an” before indirect
Line 128: one colony, not colonies
Line 146: “made of” (not “by”)
Line 187: “was measured” “w” is missing
Line 206: Have you corrected the LSD p-values manually for multiple comparisons? As stated in the GraphPad manual, the LSD test does NOT account for multiple comparisons: “Fisher's LSD test, you'll need to account for multiple comparisons when you interpret the data, since the computations themselves do not correct for multiple comparisons.” https://www.graphpad.com/guides/prism/latest/statistics/stat_fishers_lsd.htm
Which method did you choose to adjust the p-values? Just describe it here briefly (Tukey, Hot, Bonferroni, etc).
Line 217: P-value reporting: Don’t just report the p-value, also add the test statistic! This is true for all following p-values. You could also refer to a table with all the statistical results.
Line 220: “differences” not “different”
Lines 221 - 222: You could make this sentence easier to understand: “Only the Chl a concentration differed significantly between the control and the direct contact group at ambient temperature.”.
Line 227: “significant” not “significance”
Lines 228 - 229: Rather: “The interaction of temperature and algae influenced the feeding rate significantly”.
Line 231: “are shown” not “was shown”.
Lines 251 - 252: Adapt this sentence as in the example for lines 228 - 229.
Line 291: Remove “While,”.
Line 298 “suggest” instead of “suggested”.
Line 339: Not more than “significant”, more pronounced, more severe, or stronger

Figure 1 description:
You could add: “[…] with a feedback loop to adjust for the water temperature”.

Figure 2E: Why are parts of the lines solid and not dotted?

Figures 4 and 5: Description: “temperature treatment” instead of “temperatures”.

Table 1 description: Here, you missed to change peltate to peltata.

Experimental design

This is the 3rd time I am reviewing this article. I am now happy with the description of the experiment.

Validity of the findings

Check if the LSD test results were actually adjusted for multiple comparisons.

Additional comments

I am happy that this article improved and I hope that it can be published soon after some minor adjustments.

---

## Round 0.4 · accepted · Accept

Dear Authors

I am pleased to inform you that the manuscript has been accepted for publication.

The reviewer and I agree that you fulfill all the requirements.